# Hyperchloremia and Hypernatremia Decreased Microglial and Neuronal Survival during Oxygen–Glucose Deprivation/Reperfusion

**DOI:** 10.3390/biomedicines12030551

**Published:** 2024-02-29

**Authors:** Reetika Mahajan, Faheem Shehjar, Adnan I. Qureshi, Zahoor A. Shah

**Affiliations:** 1Department of Medicinal and Biological Chemistry, College of Pharmacy and Pharmaceutical Sciences, University of Toledo, Toledo, OH 43614, USA; reetika.mahajan@utoledo.edu (R.M.); faheem.shehjar@utoledo.edu (F.S.); 2Zeenat Qureshi Stroke Institutes and Department of Neurology, University of Missouri, Columbia, MO 65211, USA

**Keywords:** hyperchloremia, hypernatremia, oxygen glucose deprivation/reperfusion, nitric oxide, caspase-1/-3, human microglial cells, human neuronal cells

## Abstract

Hyperchloremia and hypernatremia are associated with higher mortality in ischemic stroke, but it remains unclear whether their influence directly contributes to ischemic injury. We investigated the impact of 0.9% sodium chloride (154 mM NaCl), 0.9% sodium acetate (167 mM CH_3_COONa), and their different combinations (3:1, 2:1, and 1:1) on microglial (HMC-3) and neuronal (differentiated SH-SY5Y) survival during oxygen–glucose deprivation/reperfusion (OGD/R). Further, we assessed the effect of hyperchloremia and hypernatremia-treated and OGD/R-induced HMC-3-conditioned media on differentiated SH-SY5Y cells under OGD/R conditions. We performed cell viability, cell toxicity, and nitric oxide (NO) release assays and studied the alteration in expression of caspase-1 and caspase-3 in different cell lines when exposed to hyperchloremia and hypernatremia. Cell survival was decreased in 0.9% NaCl, 0.9% CH_3_COONa, combinations of HMC-3 and differentiated SH-SY5Y, and differentiated SH-SY5Y cells challenged with HMC-3-conditioned media under normal and OGD/R conditions. Under OGD/R conditions, differentiated SH-SY5Y cells were less likely to survive exposure to 0.9% NaCl. Expression of caspase-1 and caspase-3 in HMC-3 and differentiated SH-SY5Y cells was altered when exposed to 0.9% NaCl, 0.9% CH_3_COONa, and their combinations. A total of 0.9% NaCl and 0.9% CH_3_COONa and their combinations decreased the NO production in HMC-3 cells under normal and OGD/R conditions. Both hypernatremia and hyperchloremia reduced the survival of HMC-3 and differentiated SH-SY5Y cells under OGD/R conditions. Based on the OGD/R in vitro model that mimics human ischemic stroke conditions, it possibly provides a link for the increased death associated with hyperchloremia or hypernatremia in stroke patients.

## 1. Introduction

Chloride is the most abundant transportable anion in all body cells, serving as the principal compensatory ion for the movement of major cations such as Na^+^, K^+^, and Ca^2+^ [1,2]. Chloride ions (Cl^−^) and sodium ions (Na^+^) are integral components in maintaining the delicate balance of ion homeostasis within the central nervous system (CNS) [3,4,5,6,7]. Their roles extend beyond electrochemical equilibrium, as they play pivotal roles in regulating neuronal and glial cell homeostasis, thus influencing fundamental processes essential for proper brain function [5,6,7]. Perturbations in Cl^−^ and Na^+^ concentrations can lead to profound consequences, ranging from neuronal excitability dysregulation to the manifestation of various genetic disorders, like neonatal seizures and epilepsy, ataxia, hyperekplexia (startle disease), and autism spectrum disorders [4,8,9,10]. Additionally, such disturbances can significantly impact the pathology following acute brain injuries, including hypoxic–ischemic encephalopathy, brain edema, and post-traumatic seizures [11,12,13,14].

Intracellular Cl^−^ is especially critical for modulating neuronal excitability by influencing postsynaptic responses to neurotransmitters such as gamma-aminobutyric acid (GABA) and glycine, key inhibitory neurotransmitters in the CNS [15,16,17]. Disruptions in Cl^−^ homeostasis, such as hyperchloremia, can alter the reversal potential of GABAergic and glycinergic neurotransmission, ultimately leading to increased neuronal excitability [18]. Similarly, intracellular Na^+^ concentration plays a vital role in maintaining cellular osmotic balance, membrane potential, and the proper functioning of ion channels and transporters [19]. Conditions such as dehydration or excessive sodium intake can result in hypernatremia, which can disrupt cellular homeostasis, alter membrane potential, and elevate neuronal excitability [20]. Hypernatremia has been associated with adverse neurological outcomes, including cerebral edema [21], cerebral impairment along with cognitive dysfunction and delirium [22,23], and myelinolysis [24]. Neurons and microglial cells maintain a finely tuned balance of extracellular Cl^−^ and Na^+^ concentrations, crucial for normal cellular function. Microglial activation is influenced by changes in ionic homeostasis (Na^+^, Ca^2+^, K^+^, H^+^, and Cl^−^), including proliferation, migration, cytokine release, and the production of reactive oxygen species [25]. Within neurons, the internal concentrations of Cl^−^ and Na^+^ contribute to establishing and modulating the resting membrane potential, influencing cellular excitability [18]. Cl^−^ and Na^+^ play pivotal roles in neuronal signaling in the extracellular environment. The concentration gradients of these ions are carefully regulated to ensure the proper transmission of signals between neurons [18]. Ischemia reduces blood flow and oxygen supply to tissues, leading to the failure of energy-dependent ion pumps, causing an influx of sodium ions and an altered chloride distribution, adversely affecting neuronal and microglial ion homeostasis and causing cytotoxic edema [26]. Exposing neuronal cells to excess NaCl reduces glucose metabolism and ATP levels, resulting in neuronal death [27]. Na^+^ and Cl^−^ disturbances are common in patients with brain injury because of the dysregulation of ion and water homeostasis. In addition, available treatments for brain injury can disturb the regulation of ions and water. These ion disturbances can lead to severe complications and adverse outcomes, including death [11,12,13,14,28]. In the last two decades, numerous studies have suggested that free radical nitric oxide (^⋅^NO) regulates the physiological responses to brain hypoxia–ischemia and reperfusion [29]. It has been found that NO has a dual role during ischemia–reperfusion [30]. However, studies have shown that salt loading in humans decreases NO release [31]. The decrease in NO release due to salt loading may negatively affect vascular health in the brain and cause cognitive impairment [32]. Given the critical role of Cl^−^ and Na^+^ in the CNS and the detrimental consequences of their dysregulation, it becomes imperative to investigate their effects under ischemic reperfusion conditions. Oxygen glucose deprivation/reperfusion (OGD/R) is a well-established experimental model employed to replicate ischemic-reperfusion conditions in vitro, offering insights into the impact of energy depletion and anoxia on various cell types, including microglial and neuronal cells.

Previous studies in human and animal experimental models have indicated that hyperchloremia and hypernatremia can exacerbate neuronal injury [18,20,22,33,34,35,36,37] and increase the chances of death or disability following ischemic stroke, intracerebral hemorrhage, and traumatic brain injury. In patients with major trauma, administration of saline post-trauma leads to hyperchloremia, which is associated with increased mortality rates [38]. In a recent clinical finding, it was observed that the occurrence of early hyperchloremia in the first 24 h was associated with a higher mortality rate in patients with severe TBI [37]. However, it remains unclear whether hyperchloremia directly causes neural injury. Identification of direct neural injury caused by hyperchloremia is important because chloride-deficient intravenous fluids are commercially available and may be better suited for patients with acute ischemic stroke. Therefore, the current study was conducted to comprehensively assess the effects of hyperchloremia and hypernatremia on the survival of microglial (HMC-3) and neuronal (differentiated SHSY-5Y) cells. Furthermore, we challenged differentiated SHSY-5Y cells with HMC-3 cell-conditioned media during OGD/R to assess the effect of hyperchloremia and hypernatremia. We investigated the impact of hyperchloremia and hypernatremia on cellular viability, cell cytotoxicity, caspase-1 and caspase-3 protein expressions, and nitric oxide release during OGD/R conditions.

## 2. Materials and Methods

### 2.1. Cell Culture

HMC-3 (human microglia) and SH-SY5Y (neuroblastoma) cell lines were purchased from the American Type Culture Collection (Rockville, MD, USA). HMC-3 cells were cultured in Eagle’s Minimum Essential Medium (EMEM) (Corning Inc., Corning, NY, USA) supplemented with 10% FBS (fetal bovine serum) (VWR) and 1% penicillin–streptomycin (Corning Inc.). SH-SY5Y cells were cultured in Dulbecco’s Modified Eagle Medium/Ham’s F12 (DMEM/F12) medium containing 10% FBS, 5% HS (horse serum), and 1% penicillin–streptomycin. For differentiation of neuroblastoma cell lines into neuronal cells, 60% confluent undifferentiated SH-SY5Y cells were treated with 10 μM retinoic acid for five days (Appendix A). Cells were incubated at 37 °C in a humidified atmosphere of 5% CO_2_. The cells were routinely passaged by trypsinization and re-seeded into 24- or 96-well plates. The medium was replaced by a fresh medium the following day, and the plates were used for further experimental treatment.

### 2.2. Treatment of HMC-3 and Differentiated SH-SY5Y Cells with NaCl and CH_3_COONa

A total of 80% confluent HMC-3 and differentiated SH-SY5Y cells were treated with different concentrations of NaCl and CH_3_COONa to study the effects of hyperchloremia and hypernatremia. Five treatments were evaluated, i.e., 0.9% NaCl, 0.9% CH_3_COONa, and different combinations, i.e., 3:1, 2:1, and 1:1 of 0.9% NaCl and 0.9% CH_3_COONa, respectively. Hereafter, these combinations will be mentioned as 3:1, 2:1, and 1:1. In another experiment, differentiated SH-SY5Y cells were exposed to media derived from or conditioned from differently treated HMC-3 cells. The concentration of 0.9% NaCl was 154 mM, and the concentration of 0.9% CH_3_COONa was 167 mM. To mimic hyperchloremia and hypernatremia conditions in cell-based assays, the concentration of Cl^+^ should be more than 110 mM, and the concentration of Na^+^ should be more than 145 mM, respectively. Thus, taking this into consideration and the normal saline concentration, we selected 154 mM NaCl and 167 mM CH_3_COONa to conduct this study. Hereafter, in this manuscript, 0.9% NaCl and 0.9% CH_3_COONa will represent hyperchloremia and hypernatremia, respectively.

### 2.3. Induction of OGD/R in HMC-3 and Differentiated SH-SY5Y Cells

To mimic ischemia-like conditions in vitro, HMC-3 and differentiated SH-SY5Y cells were washed with 1X phosphate-buffered saline to remove the residual glucose and FBS, and then cells were cultured in glucose-free balanced salt solution (140 mM NaCl, 3.5 mM KCl, 0.4 mM KH_2_PO_4_, 5 mM NaHCO_3_, 1.3 mM CaCl_2_, 1.2 mM MgSO_4_, 20 mM HEPES, pH 7.4, bubbled with 95%/5% N_2_/CO_2_) and placed in an air-tight container subsequently purged three times with 95%/5% N_2_/CO_2_. The container was placed in a 37 °C incubator for 4 h to induce OGD. After 4 h, OGD was discontinued, and the cells were transferred to the normal conditioned DMEM-F12/EMEM (reperfusion) with or without 0.9% NaCl and 0.9% CH_3_COONa in three different combinations, i.e., 3:1, 2:1, and 1:1 for 24 h. Cells exposed to normoxic/normoglycemic conditions were used as a control. 

For exposing differentiated SH-SY5Y with HMC-3 cell-conditioned media, firstly, HMC-3 cells were subjected to OGD for 4 h and then reperfused with or without media containing 0.9% NaCl and 0.9% CH_3_COONa and three different combinations, i.e., 3:1, 2:1, and 1:1, for 24 h. After 24 h, the media was collected from the OGD/R-induced HMC-3 cells and introduced to OGD-induced differentiated SH-SY5Y cells for 24 h. Cells exposed to normoxic/normoglycemic conditions were used as controls. Three replicates were used in each experiment.

### 2.4. Cell Viability Assay 

Cell viability of HMC-3 differentiated SH-SY5Y and differentiated SH-SY5Y cells exposed to HMC-3-conditioned media after treatment with 0.9% NaCl, 0.9% CH_3_COONa, and different combinations of the two was measured using CCK-8 reagent (Dojindo, Rockville, MD, USA) following the manufacturer’s instructions. Different cells were seeded into 96-well plates at a cell density of 25,000 cells/well, and the above-mentioned treatments were given. CCK-8 reagent (Dojindo) was added, and the plates were incubated for 3–4 h till there was a visible change in color. The microplate reading was taken at 450 nm (BioTek Synergy H1 Plate Reader, BioTek, Winooski, VT, USA). 

### 2.5. LDH Assay

Cell toxicity was measured by the lactate dehydrogenase (LDH) assay (Takara Bio Inc., Shiga, Japan), which assesses the membrane integrity of cells. After experimental treatment, the culture medium was collected and mixed with substrate, enzyme, and dye solutions. After 30 min of incubation in the dark, the reaction was terminated by adding a 1:10 volume of 1 N hydrochloric acid. Absorbance was measured at 490 nm and 600 nm for the reference wavelength [39]. 

### 2.6. Nitric Oxide Release

Nitric oxide production was measured using the Griess reagent kit (Biotium, Fremont, CA, USA) following the manufacturer’s protocol. The cell culture media was mixed with Griess reagent (0.5% N-(1-naphthyl) ethylenediamine dihydrochloride, 0.5% sulfanilic acid, and 2.5% phosphoric acid), followed by incubation for 30 min at room temperature, and absorbance was taken at 548 nm. 

### 2.7. Western Blotting (WB)

HMC-3, differentiated SH-SY5Y, and differentiated SH-SY5Y treated with HMC-3-conditioned media were exposed to 4 h of OGD followed by 24 h of treatment and were harvested and lysed by RIPA (radioimmunoprecipitation assay) buffer. Equivalent amounts of proteins determined by the Bradford assay were loaded onto 12% tris-SDS (Sodium Dodecyl Sulfate) gels and separated by electrophoresis. Proteins were transferred to a PVDF (polyvinylidene difluoride) membrane, blocked with 1% BSA (bovine serum albumin) in TBST (Tris-buffered saline with Tween 20), and incubated with primary antibodies in blocking buffer (anti-caspase1 (1:1000); anti-caspase3 (1:1000); anti-β-actin (1:2000); anti-β-tubulin (1:2000) (Cell Signaling Technology, Danvers, MA, USA) overnight at 4 °C, followed by secondary antibody (HRP goat anti-rabbit IgG1:10,000, Cell Signaling Technology) incubation in blocking buffer for 1 h at room temperature. Images were acquired using BioRad Chemi Doc™ XRS+ (BioRad, Hercules, CA, USA), and densitometry was analyzed using ImageJ 1.x software (developed at the National Institutes of Health (Madison, WI, USA) and the Laboratory for Optical and Computational Instrumentation, (Bethesda, MD, USA) normalized to β-actin or β-tubulin loading control.

### 2.8. Statistical Analysis

Cell viability, LDH assay, nitric oxide assay, and caspase protein expression were expressed as the mean ± SEM (standard error mean). Cell viability assays, LDH assays, nitric oxide assays, and immunoblots with three or more groups were analyzed by two-way ANOVA with Bonferroni correction to adjust for multiple comparisons between the two conditions. Statistical analyses were performed using GraphPad Prism 9 for Windows (GraphPad Software, San Diego, CA, USA). The significance level was set at * *p* < 0.05, ** *p* < 0.01, *** *p* < 0.001, and **** *p* < 0.0001.

## 3. Results

### 3.1. Effect of Hyperchloremia and Hypernatremia on Microglial Survival, Caspase-1, and Caspase-3 Expression under Normal and OGD/R Conditions

We found a highly significant reduction (*p* < 0.0001) in the cell viability in the HMC-3 cells treated with 0.9% NaCl and 0.9% CH_3_COONa compared with controls under both normal and OGD/R conditions (Figure 1a). Cell survival was lower with 0.9% NaCl than CH_3_COONa under both conditions (Figure 1a). The HMC-3 cells treated with different combinations, i.e., 3:1, 2:1, and 1:1, showed a significant reduction in cell viability compared to controls under both conditions (Figure 1a). The reduction in cell viability was further confirmed by studying cell toxicity after various treatments in cell culture media. A significant increase in cell death was found in variously treated cell media compared to control media under both normal and OGD/R conditions (Figure 1b). Further, we found that the expression levels of caspase-1 and caspase-3 were significantly changed after treating the HMC-3 cells with 0.9% NaCl, 0.9% CH_3_COONa, and different combinations both under normal as well as OGD/R conditions (Figure 1c–e). A significant increase in the expression of caspase-1 was observed in HMC-3 cells treated with a 3:1 combination compared with controls (Figure 1d). A significant increase in the expression of caspase-3 was found in all the treatments except NaCl under the OGD/R condition compared with the control group (Figure 1e). Caspase-3 activity was significantly increased in the 3:1 combination compared with the control under the OGD/R condition (Figure 1e). 

### 3.2. Effect of Hyperchloremia and Hypernatremia on Neuronal Survival and Caspase-3 Expression under Normal and OGD/R Conditions

Under normal conditions, adding 0.9% NaCl to the media significantly reduced the cell viability by more than 50% compared with the control (Figure 2a). Adding 0.9% CH_3_COONa resulted in higher cell viability than 0.9% NaCl (Figure 2a). Around a 50% reduction in cell viability (compared with the control) was identified with different combinations, i.e., 3:1, 2:1, and 1:1 (Figure 2a). Likewise, a significant decrease in cell viability was observed in OGD-induced differentiated SH-SY5Y cells reperfused with or without different treatment media (Figure 2a). Reperfusion of OGD-induced differentiated SH-SY5Y cells reduced the cell viability to around 30% compared with control (Figure 2a). Further, reperfusion with different media showed too few viable cells compared with the control (Figure 2a). The LDH assay demonstrated a significant increase in cell death as compared with control in the cell media collected from different treatments under both normal and OGD/R conditions (Figure 2b). Moreover, the expression level of caspase-3 protein was significantly changed after treating the differentiated SH-SY5Y cells with 0.9% NaCl, 0.9% CH_3_COONa, and different combinations of both under normal as well as OGD/R conditions (Figure 2c,d). A significant increase in the expression of caspase-3 was observed in differentiated SH-SY5Y cells treated with CH_3_COONa and in a 3:1 combination compared with controls under OGD/R (Figure 2d), whereas a significant decrease in the expression of caspase-3 was found in 0.9% NaCl and in combinations 2:1 and 1:1 under OGD/R conditions compared with controls (Figure 2d).

### 3.3. Effect of Hypernatremia and Hyperchloremia on Cell Viability and Caspase Protein Expression of Neuronal Cells Exposed to HMC-3 OGD/R-Conditioned Media

Exposing the differentiated SH-SY5Y cells to HMC-3 OGD/R-conditioned media significantly reduced cell viability to 25% compared with the control. Differentiated SH-SY5Y cells were subjected to 4 h of OGD and reperfused with or without different media collected from OGD/R-subjected HMC-3 cells (Figure 3a). Under normal conditions, however, all treatments showed approximately 60% cell viability as compared to the control (Figure 3a). Further, the results were confirmed by evaluating the cell cytotoxicity in cells exposed to normal and OGD/R-conditioned media (Figure 3b). The effect of 0.9% NaCl, 0.9% CH_3_COONa, and different combinations of both on the expression of two different caspases, i.e., caspase-1 and caspase-3, was investigated under normal and OGD/R-conditioned media (Figure 3c). Caspase-1 expression level was significantly decreased in the 2:1 and 1:1 combination compared with control under normal conditions (Figure 3d). Meanwhile, under the OGD/R condition, the 2:1 and 1:1 combination significantly increased the caspase-1 protein expression compared with the OGD/R control (Figure 3d). A difference in the expression pattern of the caspase-3 protein between normal and OGD/R was observed (Figure 3e). Under normal conditions, caspase-3 activity significantly increased in different treatments except 0.9% NaCl compared with the control (Figure 3e); however, a significant decrease in the expression of caspase-3 was found in 0.9% CH_3_COONa-treated and different combination-treated conditioned cells compared with the control under the OGD/R condition (Figure 3e).

### 3.4. Effect of Increased NaCl and CH_3_COONa on Nitric Oxide Production in HMC-3 and Differentiated SH-SY5Y Cells Challenged with HMC-3-Conditioned Media

A significant decrease in nitric oxide production with exposure to 0.9% NaCl, 0.9% CH_3_COONa, and different combinations of both in the cellular media of HMC-3 cells under normal and OGD/R conditions was found (Figure 4a,b). However, differentiated SH-SY5Y cells treated with HMC-3-conditioned media showed a significant increase in nitric oxide production in 0.9% CH_3_COONa compared to other treatments under normal conditions. Meanwhile, under OGD/R conditions, a 3:1 and 2:1 combination showed a significant increase in nitric oxide release compared to other treatments (Figure 4c,d). 

## 4. Discussion

The brain has different cell types, but the research has mainly focused on microglial, neuronal, and astrocytes [40]. In our research, we chose neuronal and microglial cells. The exclusion of astrocytes in our research was a deliberate methodological choice. While astrocytes play a crucial role in brain homeostasis, we focused on microglial and neuronal cells to address a targeted research question and align with established methodologies. This selective exclusion allows for a more precise examination of the effects of sodium imbalances on microglia and neurons in the context of OGD/R, contributing to a clearer understanding of the direct interactions between these cell types without the confounding influence of astrocytic responses. Such focused investigations contribute to broader scientific knowledge and enhance the specificity of our findings concerning the chosen cellular components. Our investigation revealed that both 0.9% NaCl and 0.9% CH_3_COONa treatments significantly reduce the viability of microglial and neuronal cells, and the neuronal cells were challenged with conditioned media from microglial media under both normal and OGD/R conditions. Intriguingly, 0.9% NaCl exhibited greater toxicity than 0.9% CH_3_COONa in both scenarios, highlighting the possibility of distinct cellular responses to different sodium salts and indicating that ion imbalance may have a more pronounced impact on the viability of both microglial and neuronal cells, as well as conditioned microglial media exposed neuronal cells. Additionally, various combinations of NaCl and CH_3_COONa (3:1, 2:1, and 1:1) led to a notable decrease in cell viability compared to the control groups under both conditions. These findings emphasize the vulnerability of microglial, neuronal, and neuronal cells challenged with conditioned media microglial cells to hyperchloremia and hypernatremia, suggesting that electrolyte imbalances could contribute to neuroinflammation, compromise cell survival, and increase the cerebral perfusion pressure during ischemic stroke [41]. Under OGD/R conditions, both microglial and neuronal cells and neuronal cells exposed to conditioned microglial media proved even more susceptible to the toxic effects of excess NaCl and CH_3_COONa. OGD alone caused a drastic reduction in cell viability, and when reperfused with different treatments or conditioned media, cell viability decreased further, underscoring the exacerbating role of electrolyte imbalances in post-ischemic-like conditions. Challenging neuronal cells with OGD-induced microglial-conditioned media revealed additional layers of complexity in cellular responses. The substantial decrease in differentiated SH-SY5Y cell viability, especially under OGD/R conditions, emphasizes the potentially detrimental effects of conditioned microglial cell media on neuronal viability. This observation aligns with the growing understanding of the critical role of microglia in neuronal support and neuroinflammation.

Moreover, assessing cell toxicity in culture media mirrored the trends in cell viability for both microglial and neuronal cells and neurons exposed to conditioned microglial media under normal and OGD/R conditions, confirming the observed reductions in cell viability. In the presence of excess NaCl and CH_3_COONa, a significant increase in cell death was observed in the cells subjected to different treatments. This observation highlights the detrimental effects of excess NaCl and CH_3_COONa on microglial cells, neuronal cells, and neuronal cells that were exposed to conditioned microglial cell media under OGD/R conditions. Furthermore, based on the fact that LDH release is a marker for detecting cell necrosis [42], it can be concluded from our results that hyperchloremia leads to cell necrosis as the LDH release was higher in the NaCl treatment as compared to all other treatments under both normal and OGD/R conditions. Thus, our results suggest that hyperchloremia is more detrimental than hypernatremia under normal and OGD/R conditions for microglial, neuronal, and microglial-conditioned media-exposed neuronal cells. The observed cellular damage underlines the importance of maintaining electrolyte balance for normal cell function and survival. 

As caspases have a key role in apoptosis, our cell viability/cytotoxicity results suggest that increased concentrations of NaCl and CH_3_COONa lead to cell death by apoptosis or necrosis. So, we decided to check whether excess NaCl and CH_3_COONa caused cell death via caspase-dependent pathways in microglial and neuronal cells under OGD/R. In microglial cells, caspase-1 expression was significantly increased in the 3:1 combination, suggesting potential involvement in the inflammatory response induced by Na^+^ and Cl^−^ ions. Conversely, caspase-3 expression was increased in various treatments under normal conditions, except 0.9% NaCl. However, under the OGD/R condition, caspase-3 expression was significantly higher in the 0.9% CH_3_COONa and in different combinations, i.e., 3:1 and 2:1, as compared to the control, suggesting that reperfusion of OGD-subjected microglial cells with hypernatremia can lead to cell death via a caspase-3 dependent pathway. A significant decrease in caspase-3 and caspase-1 expression in the presence of 0.9% NaCl suggested that hyperchloremia might follow caspase-independent cell death pathways in microglial cells under OGD/R conditions. Further molecular studies are needed to validate these results. Notably, the 0.9% CH_3_COONa and the 3:1 combination exhibited increased caspase-3 activity under OGD/R, implying that supplementation of excess Na^+^ to neuronal cells contributes to caspase-3-dependent cell death. Furthermore, previous reports published from our lab suggest that sustained microglial activation is deleterious for the brain, as activated microglia release pro-inflammatory cytokines in the media, which can either lead to inflammasome formation or cell death by caspase-dependent or independent pathways [43,44,45]. Moreover, we challenged the differentiated SH-SY5Y cells with HMC-3-conditioned media with the thought that OGD/R-induced HMC-3 cells treated with 0.9% NaCl and 0.9% CH_3_COONa and their different combinations might secrete pro-inflammatory cytokines in the cellular media, and when these media are added to differentiated SH-SY5Y cells, these pro-inflammatory cytokines would affect the caspase activity of the neuronal cells under hypoxia. Interestingly, we found neuronal cells exposed to conditioned microglial media yielded complex results in caspase protein expression. Caspase-1 expression decreased in the presence of a 2:1 and 1:1 combination under normal conditions but increased under OGD/R. This suggests a potential role in exacerbating neuronal cell death during ischemic-reperfusion-like conditions. Conversely, caspase-3 protein exhibited different expression patterns, with increased expression under normal conditions in various treatments except 0.9% NaCl and decreased expression in 0.9% CH_3_COONa-treated and combination-treated conditioned cells under OGD/R. Increased caspase-1 and decreased caspase-3 expression under OGD/R with 0.9% NaCl indicate that excess NaCl might cause cell death via pyroptosis in neuronal cells exposed to conditioned microglial media in post-ischemic conditions. Earlier studies conducted in different cell lines also suggested that salts like NaCl and KCl inhibit caspase-1 activation [46,47]. Caspases are an evolutionary conserved family of cysteine proteases that mediate regulated cell death, including apoptosis and pyroptosis [48]. Caspase-1 and caspase-3, critical players in apoptosis and inflammation, exhibited significant alterations in their expression and activity in response to an increase in ion concentration [49]. Most of the caspases are either initiators (caspases 2, 8, 9, and 10) or executioners (caspases 3, 6, and 7) of apoptosis [50,51,52]. Inflammatory caspases like caspase-1, caspase-4, and caspase-5 regulate pyroptosis in humans [51,52,53]. It has been reported that caspase-dysregulated activation plays diverse but important roles in multiple neurodegenerative diseases, including brain injury and neuroinflammation [54]. The differential modulation of caspase-1 and caspase-3 expression under different conditions further highlights the intricate interplay between microglial and neuronal cells regulating apoptotic pathways in response to electrolyte imbalances. This information may be relevant to understanding the cellular processes involved in diseases associated with electrolyte imbalances.

NO, a vital signaling molecule, plays diverse roles in cellular processes [55]. Our study demonstrated a significant decrease in NO production in HMC-3 microglial cells exposed to elevated NaCl and CH_3_COONa concentrations under both normal and OGD/R conditions. This reduction in NO production may influence the regulation of neuroinflammatory responses, as NO is integral to modulating neuronal cell functions. Moreover, OGD/R-induced SH-SY5Y cells, when exposed to HMC-3-conditioned media, showed different results, suggesting that the interactions between microglial and neuronal cells in the presence of excess NaCl and CH_3_COONa can modulate NO signaling, with potential implications for neuroinflammation and neuronal cell viability post-ischemia. The freely diffusible gaseous compound NO is an important messenger in many organ systems, particularly the CNS [56]. Earlier studies also suggested high salt intake can reduce NO production in different cell types [31,57]. However, exposure to a 3:1 and 1:1 combination of NaCl and CH_3_COONa-treated microglial media in neuronal cells showed an increase in nitric oxide (NO) production in neuronal cells. Earlier studies revealed that increased Ca^2+^ ion concentration in astrocytes and neuronal cells enhanced NO production, which can contribute significantly to the toxic effects of OGD and lead to cell death [58,59]. Here, the increase in NO production in neuronal cells treated with microglial media suggested that hyperchloremia and hypernatremia-treated neuronal cell-challenged microglial media lead to more cell death after OGD/R. However, further molecular investigation is needed to support the change in NO production in microglial media-challenged neuronal cells.

While our study provides valuable insights into the effects of hyperchloremia and hypernatremia on microglial and neuronal cells under OGD/R conditions, the controlled environment of in vitro experiments may not fully capture the complexity of the in vivo neural microenvironment. In vitro models lack the intricate interplay of systemic factors, cellular interactions, and dynamic physiological responses in living organisms. Therefore, extrapolating our findings to the in vivo scenario should be approached with caution. Future studies employing complementary in vivo models are warranted to validate and extend our observations, providing a more comprehensive understanding of the implications of ion homeostasis and nitric oxide regulation in ischemic reperfusion injury. This acknowledgment of the limitations reinforces the need for a multidimensional approach to fully elucidate the complexity of neural responses to pathological conditions.

### Clinical Implications

A total of 0.9% NaCl is the most commonly used intravenous fluid in patients with acute ischemic stroke. Hyperchloremia can occur due to the high chloride load in 0.9% NaCl, which has been associated with higher rates of death or disability in patients with ischemic stroke [36]. Our study provides insights into the potential mechanism of an increase in neuronal injury associated with high chloride in the setting of ischemia. In normal circumstances, the excess serum chloride is unlikely to move to extracellular spaces in the brain due to the intact blood–brain barrier (BBB). However, the BBB is disrupted in the ischemic region, allowing chloride to increase in the extracellular spaces, affecting cell viability during ischemia [60,61]. Our findings may have implications for selecting intravenous fluids, emphasizing intravenous fluids with a lower chloride load. An ongoing clinical trial is comparing intravenous 0.9% NaCl with various combinations of intravenous NaCl:CH_3_COONa combinations in patients with acute stroke (NCT05869565) (https://clinicaltrials.gov/study/NCT05869565, accessed on 23 November 2023). The combination of findings from both our current report and the clinical study will assist in determining the best intravenous fluid for patients with acute ischemic stroke. 

## 5. Conclusions

Our study demonstrates that excess NaCl and CH_3_COONa can significantly impact the viability, cytotoxicity, caspase protein expression, and NO production of microglial and neuronal cells. The findings reveal a marked reduction in cell viability in both HMC-3 (microglial) and differentiated SH-SY5Y (neuronal cells) when exposed to 0.9% NaCl, 0.9% CH_3_COONa, and various combinations of these solutions under normal and OGD/R conditions. Additionally, the altered expression of caspase-1 and caspase-3 proteins in both cell types suggests a potential involvement of apoptosis pathways in the observed cellular responses. The observed decrease in nitric oxide production in HMC-3 cells under hyperchloremic and hypernatremic conditions, while contrasting with an increase in differentiated SH-SY5Y cells conditioned with HMC-3 media, highlights the intricate and cell-type-specific responses to ion disturbances. These findings contribute to our understanding of the complex interplay between ion homeostasis and cellular responses in the context of ischemic reperfusion injury. Understanding how these electrolyte imbalances affect cell viability, cytotoxicity, caspase protein expression, and NO production may have clinical implications, mainly when electrolyte imbalances are prevalent. However, it is crucial to acknowledge the limitations of the in vitro experimental setup, which may not fully capture the complexity of in vivo neural responses. Future studies employing complementary in vivo models are warranted to validate and extend our observations. Despite this limitation, our study provides valuable insights into the potential mechanistic basis for the increased mortality and disability associated with hyperchloremia and hypernatremia in OGD/R conditions.

## Figures and Tables

**Figure 1 biomedicines-12-00551-f001:**
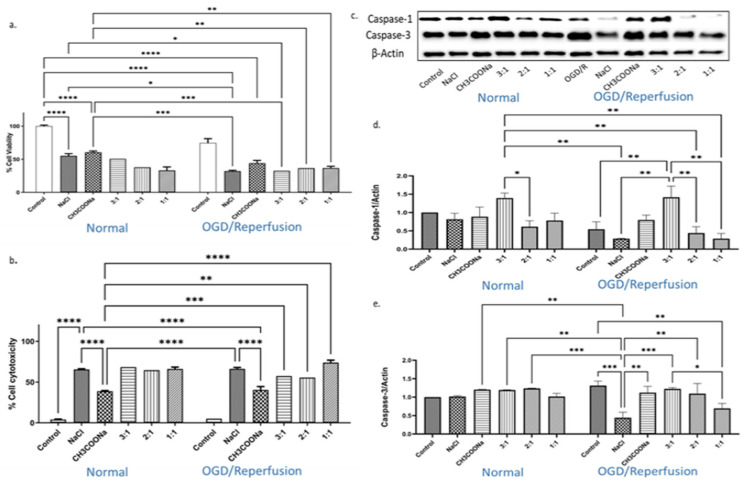
The impact of hyperchloremia and hypernatremia on cell viability, cell toxicity, and caspase-1 and -3 expression in hypoxia-induced HMC-3 cells. (**a**) Cell viability (*n* = 4); (**b**) cell toxicity (*n* = 3) of HMC-3 cells treated with 0.9% NaCl, 0.9% CH_3_COONa, and three different combinations (3:1, 2:1, and 1:1) of 0.9% NaCl and 0.9% CH_3_COONa, respectively, under normal and OGD/R induction. (**c**) The expression levels of caspase-1 and -3 were analyzed by WB. (**d**,**e**) The relative quantification of caspase-1 and caspase-3 proteins under normal and OGD/R conditions. The blots (*n* = 3) were quantified by ImageJ software and normalized with β-actin. The statistical significance was determined by a two-way ANOVA, and significant differences (*p* < 0.05, *p* < 0.01, *p* < 0.001, and *p* < 0.0001) are symbolized as *, **, ***, and ****, respectively.

**Figure 2 biomedicines-12-00551-f002:**
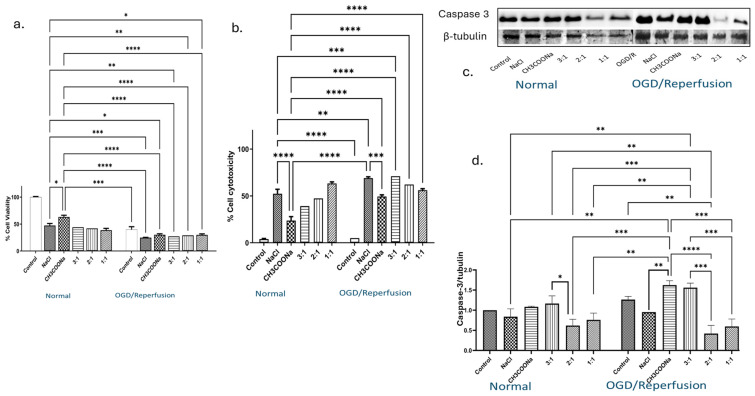
The impact of hyperchloremia and hypernatremia on caspase-3 expression and cell viability in normal and OGD/R-induced differentiated SH-SY5Y cells. (**a**) Cell viability (*n* = 4), (**b**) Cell cytotoxicity (*n* = 3) of differentiated SH-SY5Y cells exposed to 0.9% NaCl, 0.9% CH_3_COONa, and three different NaCl:CH_3_COONa ratios (3:1, 2:1, and 1:1) under both normal and OGD/R conditions, compared to the control. (**c**) Caspase-3 expression levels were analyzed by WB. (**d**) Relative quantification of caspase-3 protein levels was conducted under normal and OGD/R conditions. Quantification of blots (*n* = 3) was performed using ImageJ software and normalized to β-tubulin. Statistical significance was determined through two-way ANOVA, with significance levels denoted as follows: *, *p* < 0.05; **, *p* < 0.01; ***, *p* < 0.001; ****, *p* < 0.0001.

**Figure 3 biomedicines-12-00551-f003:**
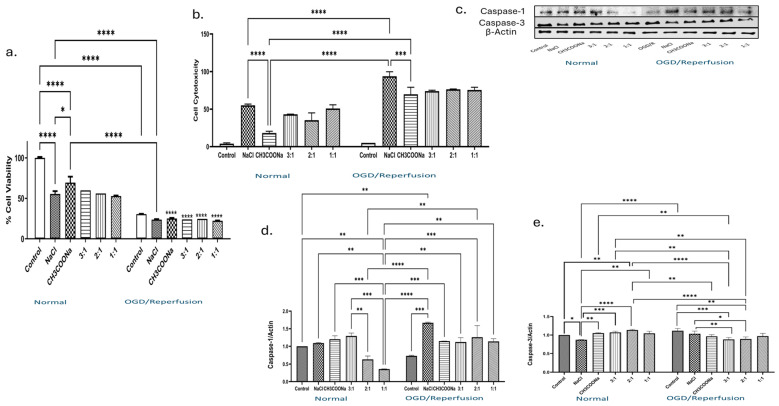
The effect of 0.9% NaCl, 0.9% CH_3_COONa, and different combinations of 0.9% NaCl and 0.9% CH_3_COONa treatment on cell viability, cell cytotoxicity, and caspase expression in OGD/R-induced differentiated SH-SY5Y conditioned with OGD/R-induced HMC-3 media. (**a**) Cell viability (*n* = 4) and (**b**) cell cytotoxicity (*n* = 3) of treated differentiated SH-SY5Y cells were further exposed to conditioned HMC-3 cell media for 24 h under normal and OGD/R conditions. (**c**) The expression levels of caspase-1 and -3 were analyzed by WB. (**d**,**e**) Different treatments of 0.9% NaCl and 0.9% CH_3_COONa significantly changed the caspase-1 and -3 expression in challenged SHSY-5Y cells with HMC-3-conditioned media as compared to the control. The blots (*n* = 3) were quantified with ImageJ software and normalized with β-actin. The significance was determined by a two-way ANOVA, and significant differences (*p* < 0.05, *p* < 0.01, *p* < 0.001, and *p* < 0.0001) are symbolized as *, **, ***, and ****, respectively.

**Figure 4 biomedicines-12-00551-f004:**
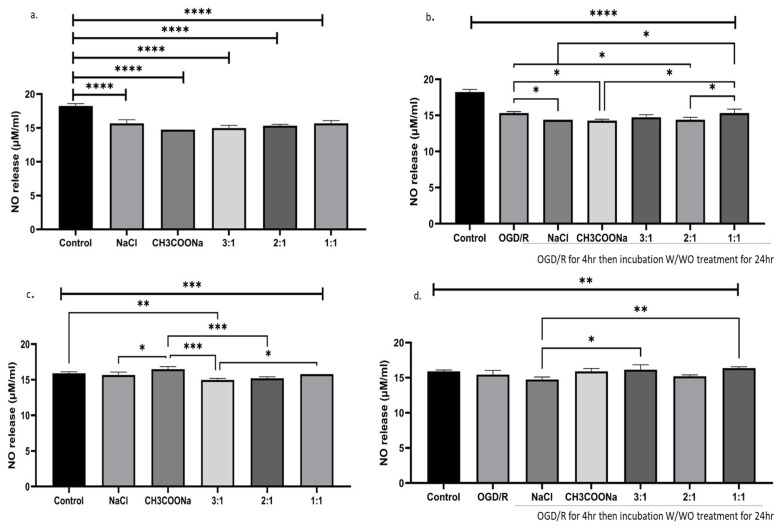
Impact of 0.9% NaCl, 0.9% CH_3_COONa, and various combinations of these treatments on nitric oxide (NO) release in HMC-3 and SHSY-5Y cells exposed to HMC-3-conditioned media under normal and OGD/R conditions (*n* = 3). (**a**,**b**) A significant decrease in the release of NO in various treatments as compared to control under normal and OGD/R in HMC-3 cells. (**c**,**d**) There was a significant change in NO release in various treatments in SHSY-5Y cells exposed to HMC-3-conditioned media compared to the control under normal and OGD/R conditions. The significance was determined by a two-way ANOVA, and significant differences (*p* < 0.05, *p* < 0.01, *p* < 0.001, and *p* < 0.0001) are symbolized as *, **, ***, and ****, respectively.

## Data Availability

All the data needed to assess the conclusions in the paper are exhibited in the paper.

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
