# Peer review of "Hyperchloremia and Hypernatremia Decreased Microglial and Neuronal Survival during Oxygen–Glucose Deprivation/Reperfusion"

_biomedicines, 2024, doi:10.3390/biomedicines12030551_

Round 1

Reviewer 1 Report (Previous Reviewer 1)

Comments and Suggestions for Authors

Reviewer comments

The present research article shows the effect of hyperchloremia and hypernatremia on microglia and neuronal cell viability in case of deprivation of glucose. The study outcome demonstrates that excess hyperchloremia and hypernatremia can significantly impact the caspase protein expression, and NO production leads to a decrease in microglial and neuronal cell growth and viability.

Recommendation

1.      The paper is scientifically sound, well-planned, and written in an organized manner.

2.      The research objective and targets of work are original and relevant to the field. The outcomes of the study are discussed briefly.

3.      Revised MS has updated all the previous comments.

4.      References are appropriate.

Author Response

We thank this reviewer for positive feedback.

Reviewer 2 Report (Previous Reviewer 2)

Comments and Suggestions for Authors

1) The abstract must be rewritten. There are some difficult to read sentences. For example - We performed cell viability, cell toxicity, and nitric oxide (NO) release??? The word We performed cell viability, cell toxicity, and nitric oxide (NO) release tests may be missing. In addition, Our findings provide a mechanistic basis for the increase in death or disability associated with hyperchloremia or hypernatremia in post-ischemic stroke patients. This expression is not convincing. The results were obtained in an in vitro system and their confident transfer to the human body is not appropriate even in the form of an assumption and the purpose of the work. In addition, the authors rightly indicate the limits of their work in the conclusions

2) For differentiation of neuroblastoma cell lines into neuronal cells, 60% confluent. Why did you use 60% confluence? With such confluency, proliferation still occurs, at a minimum, chromatin condensation is observed and viability tests may give an error. Authors must clearly justify the use of such confluent cultures.

Author Response

Response to Comment 1: 

The abstract has been rewritten as suggested.

Response to Comment 2:

Based on the literature, a 60% confluency threshold was chosen for differentiating SH-SY5Y human neuroblastoma cells. Several research articles advocate for differentiation within the range of 60-80% confluency using 10μM retinoic acid. Additionally, studies have demonstrated that retinoic acid treatment induces SH-SY5Y cells to enter the post-mitotic phase, significantly reducing proliferation.

[“Substrate stiffness effects on SH-SY5Y: The dichotomy of morphology and neuronal behavior” Alp Ozgun, Fatma Zehra Erkoc-Biradlı, Osman Bulut, Bora Garipcan https://doi.org/10.1002/jbm.b.34684]

(Ehrich et al., 1997; Ehrich et al., 2011; Emerick et al., 2012; Emerick et al., 2015). 

Ehrich, M., Correll, L., Veronesi, B., 1997. Acetylcholinesterase and neuropathy target esterase inhibitions in neuroblastoma cells to distinguish organophosphorus compounds causing acute and delayed neurotoxicity. Fundam. Appl. Toxicol. 38, 55–63.

Ehrich, M., Van Tassell, R., Li, Y., Zhou, Z., Kepley, C.L., 2011. Fullerene antioxidants decrease organophosphate-induced acetylcholinesterase inhibition in vitro. Toxicol. Vitr. 25, 301–307.

Emerick, G.L., DeOliveira, G.H., Oliveira, R.V., Ehrich, M., 2012. Comparative in vitro study of the inhibition of human and hen esterases by methamidophos enantiomers. Toxicology 292, 145–150.

Emerick, G.L., Fernandes, L.S., de Paula, E.S., Barbosa, F., Guinaim dos Santos, N.A., dos Santos, A.C., 2015. In vitro study of the neuropathic potential of the organophosphorus compounds fenamiphos and profenofos: Comparison with mipafox and paraoxon. Toxicol. Vitr. 29, 1079–1087.

Reviewer 3 Report (Previous Reviewer 3)

Comments and Suggestions for Authors

The authors have addressed all my comments. 

Author Response

We are thankful to the reviewer for approving our revisions. 

Reviewer 4 Report (Previous Reviewer 4)

Comments and Suggestions for Authors

Congratulations. The manuscript is now ready for publication.

Author Response

We thank the reviewer for approving our article.

This manuscript is a resubmission of an earlier submission. The following is a list of the peer review reports and author responses from that submission.

Round 1

Reviewer 1 Report

Comments and Suggestions for Authors

Reviewer comments

The present research article shows the effect of hyperchloremia and hypernatremia on microglia and neuronal cell viability in case of ischemic injury. The study outcome demonstrates that excess hyperchloremia and hypernatremia can significantly impact the caspase protein expression, and NO production leads to a decrease in microglial and neuronal cell growth and viability.

The paper is scientifically sound, well-planned, and written in an organized manner.

Recommendation

1.      The research objective and targets of work are original and relevant to the field. The outcomes of the study are discussed briefly.

2.      References are appropriate.

Scientific comments

1.      Is LDH assay and NO assay performed using any previously published protocol? If yes, cite that paper here.

2.      Provide a reference for a statement from lines 67-70.

3.      Discuss the 'Hyperchloremia is associated with 30-day mortality in major trauma patients: a retrospective observational study' paper in the introduction or discussion section.

4.      Discuss the paper 'https://doi.org/10.1016/j.neuro.2016.03.005' in the second paragraph of the introduction section.

5.      In Figure 2, the impact of hyperchloremia and hypernatremia was observed for caspase-3 expression but not for caspase-1 however figure 3 shows results for both caspase-3 and 1, why?

6.      One paper published by one of the co-authors and the team to investigate the ‘effect of the occurrence of early hyperchloremia on death or severe disability at 180 days in patients with severe traumatic brain injury (TBI)’, discuss this paper's outcomes with present outcomes.

7.      Mention the major limitation of in vitro experiment outcomes in the discussion or the conclusion section.

Minor comments

1.      In Line 20, delete one extra C from the caspase.

2.      Line 30-36, delete how to use this template paragraph.

3.      Check references, in some references, years are not in bold font. Correct all.

Author Response

The present research article shows the effect of hyperchloremia and hypernatremia on microglia and neuronal cell viability in case of ischemic injury. The study outcome demonstrates that excess hyperchloremia and hypernatremia can significantly impact the caspase protein expression, and NO production leads to a decrease in microglial and neuronal cell growth and viability.

The paper is scientifically sound, well-planned, and written in an organized manner.

We thank the reviewer for the positive feedback.

Recommendation

  1. Is LDH assay and NO assay performed using any previously published protocol? If yes, cite that paper here.

Response: Suggestion incorporated in the revised manuscript.

  1. Provide a reference for a statement from lines 67-70.

Response: As suggested, a reference for the statement has been added.

  1. Discuss the 'Hyperchloremia is associated with 30-day mortality in major trauma patients: a retrospective observational study' paper in the introduction or discussion section.

Response: This paper has been discussed in the introduction section, lines 97-99.

  1. Discuss the paper 'https://doi.org/10.1016/j.neuro.2016.03.005' in the second paragraph of the introduction section.

Response: Suggestion incorporated in lines 79-81.

  1. In Figure 2, the impact of hyperchloremia and hypernatremia was observed for caspase-3 expression but not for caspase-1 however figure 3 shows results for both caspase-3 and 1, why?

Response: We examined the expression of both caspases in all cell lines and consistently observed their expression. However, in SH-SY5Y cells, there was no detectable expression for caspase-1. Despite repeating the experiment three times, we did not observe caspase-1 expression, even in the control sample. Consequently, for SH-SY5Y cells, we have presented the expression data for caspase-3 only.

  1. One paper published by one of the co-authors and the team to investigate the ‘effect of the occurrence of early hyperchloremia on death or severe disability at 180 days in patients with severe traumatic brain injury (TBI)’, discuss this paper's outcomes with present outcomes.

Response: As suggested, this paper has been discussed.

  1. Mention the major limitation of in vitro experiment outcomes in the discussion or the conclusion section.

Response: As suggested major limitation of in vitro experiment has been added in the discussion section.

Minor comments

  1. In Line 20, delete one extra C from the caspase.

Response: This has been deleted.

  1. Line 30-36, delete how to use this template paragraph.

Response: This has been deleted.

  1. Check references, in some references, years are not in bold font. Correct all.

Response: References have been corrected as per the format.

Reviewer 2 Report

Comments and Suggestions for Authors

1) The abstract must be checked for errors. It is necessary to rewrite the abstract, since it should more clearly reflect the results obtained, taking into account the arsenal of methods used.

2) 0. How to Use This Template. It's too much.

3) Therefore, in the current study, we carried out a comprehensive assessment of the effects of hyperchloremia and hypernatremia on the survival of microglial (HMC-3), neuronal (SHSY-5Y) and SHSY-5Y cells challenged with HMC-3 cells conditioned media during 80 OGD/R. This is a very difficult sentence to read. Needs rephrasing.

4) From the introduction it is not at all clear why the authors decided to study the effects of hyperchloremia and hypernatremia on nitric oxide release, as well as on cell survival. The introduction shows the importance of these processes on the functioning of neuronal networks, but does not clearly indicate the known pathways of cell death activated by hyperchloremia and hypernatremia. At the same time, the activation of cell death pathways in OGD/R is well known.

5) SH-SY5Y – cannot be considered a neuronal cell line. This is a cancerous brain cell line of neuronal origin. It is necessary to clearly justify why microglial cells and cancer cells are compared, especially the effects of a conditioned environment. This is my key remark, which does not allow me to accept this work for publication and further consideration.

6) Figure 1 is difficult to read. Needs improvement in quality and resolution

7) 3.5. Effect of increased NaCl and CH3COONa on nitric oxide production in HMC-3 and cocultured SH-SY5Y cells with HMC-3 media: A completely unclear term. Co-culture is possible when two types of cells are used in one culture. In this case, do you mean pre-incubation?

Comments on the Quality of English Language

There are a lot of difficult to read sentences. necessary

Author Response

  • The abstract must be checked for errors. It is necessary to rewrite the abstract, since it should more clearly reflect the results obtained, taking into account the arsenal of methods used.

Response: As suggested, the abstract has been rewritten.

  • How to Use This Template. It's too much.

Response: We apologize for this mistake. This has been removed from the revised manuscript.

  • Therefore, in the current study, we carried out a comprehensive assessment of the effects of hyperchloremia and hypernatremia on the survival of microglial (HMC-3), neuronal (SHSY-5Y) and SHSY-5Y cells challenged with HMC-3 cells conditioned media during 80 OGD/R. This is a very difficult sentence to read. Needs rephrasing.

Response: This sentence has been rephrased in the revised manuscript.

  • From the introduction it is not at all clear why the authors decided to study the effects of hyperchloremia and hypernatremia on nitric oxide release, as well as on cell survival. The introduction shows the importance of these processes on the functioning of neuronal networks but does not clearly indicate the known pathways of cell death activated by hyperchloremia and hypernatremia. At the same time, the activation of cell death pathways in OGD/R is well known.

Response: The introduction has been modified and the details about nitric oxide release and cell survival have been added.

  • SH-SY5Y – cannot be considered a neuronal cell line. This is a cancerous brain cell line of neuronal origin. It is necessary to clearly justify why microglial cells and cancer cells are compared, especially the effects of a conditioned environment. This is my key remark, which does not allow me to accept this work for publication and further consideration.

Response: Yes, we agree with the reviewer’s comment that SH-SY5Y is thrice cloned subline of the neuroblastoma cell line SK-N-SH. However, SH-SY5Y can be differentiated into neuron-like cells by stimulation with retinoic acid. (Voogd, E.J.H.F., Frega, M. & Hofmeijer, J. Neuronal Responses to Ischemia: Scoping Review of Insights from Human-Derived In Vitro Models. Cell Mol Neurobiol 43, 3137–3160 (2023). https://doi.org/10.1007/s10571-023-01368-y). Various experimental studies addressed responses to ischemia or hypoxia in human-derived cell models. These consist of non-neuronal and neuronal models. The most used human in vitro model to study neuronal responses to ischemia or hypoxia is the neuroblastoma-derived SH-SY5Y cell model (Liu et al. 2018). This cell model is based on a cancerous cell line with the corresponding genetic characteristics, which may affect responses to ischemia or hypoxia and treatment strategies (Biedler et al. 1973, 1978). SH-SY5Y cells can, in principle, be differentiated into neuron-like cells upon stimulation (Shipley et al. 2016). However, most of the research with SH-SY5Y cell models is conducted in undifferentiated SH-SY5Y cells. Only a small proportion of the research made use of protocols to differentiate SH-SY5Y cells into neuron-like cells. Here in our study taking into consideration that these cell lines are neuroblastoma, we first differentiated the cell line into neurons with retinoic acid and then used the differentiated cell lines to study the effect of NaCl and CH3COONa. The details about the differentiation have been added to the material method section.

  • Figure 1 is difficult to read. Needs improvement in quality and resolution

Response: The figure has been modified.

7) 3.5. Effect of increased NaCl and CH3COONa on nitric oxide production in HMC-3 and cocultured SH-SY5Y cells with HMC-3 media: A completely unclear term. Co-culture is possible when two types of cells are used in one culture. In this case, do you mean pre-incubation?

Response: The sentence has been changed to “Effect of increased NaCl and CH3COONa on nitric oxide production in HMC-3 and SH-SY5Y cells challenged with HMC-3 conditioned media”.

Comments on the Quality of English Language

There are a lot of difficult to read sentences. Necessary

Response: Some of the hard-to-understand sentences have been rephrased.

Reviewer 3 Report

Comments and Suggestions for Authors

The submitted manuscript describes the effect of excessive Na+ and Cl- concentrations on the viability of HMC-3 and SH-SY5Y cells in ischemia-like conditions. The study has numerous serious methodological and conceptual flaws and cannot be recommended for publication in the current version. The comments are listed below. 

- The information about the extracellular and internal Cl- and Na+ concentration in neurons and microglia cells should be included in the introduction. The alteration of these concentrations in pathological conditions, especially ischemia, should also be added to the text.

- It is well known that the number of astrocytes in the brain is also significant. For some brain regions, the neurons to astrocytes ratio exceeds 1:2. Why did the authors ignore the astrocytes in their study?

- Neuroblastoma cells cannot be considered as neurons due to the changed profile of metabolism and membrane expression of different receptors and channels. Alterations of ion homeostasis, especially sodium, result in depolarization of neurons and excitotoxicity caused by excessive glutamate release in neuronal networks. These effects cannot be observed in the case of neuroblastoma. Moreover, the vulnerability of neurons and neuroblastoma to ischemia-like conditions significantly differs. Thus, neuroblastoma is an inappropriate model for studying the effects of ion homeostasis perturbations on neuronal survival in this case.

- The authors describe different ratios of 0,9% sodium chloride or sodium acetate, but which is the final concentration of Na+ or Cl- in the solutions? The description of this issue in the materials and methods section needs to be more accurate.

- "cells were cultured in glucose-free balanced salt solution". Which solution was used exactly? Using PBS or other Ca2+-free medium for long-term incubation can result in cell detachment, especially in the case of neuroblastoma cell lines. The full composition of the solutions, except traditionally used ones like DMEM or PBS, must be described. Moreover, the pH of the solutions and the details about pH adjustment and maintenance (used pH buffers) should be described. 

- The authors write that they used NaCl to model hyperchloremia-like conditions—however, dissociation of the NaCl results in an increase of Na+ and Cl- concentration simultaneously. The authors should have used other chlorides (choline chloride, for instance) to increase Cl- concentration. 

- What was the rationale for using the HMC-3-conditioned medium?

- Why did the authors estimate the effect of changes in extracellular ion concentration on NO production? 

- The number of experiments for each diagram must be indicated in the figure legend. The significance of some differences, especially in Fig.4, raises some concerns. Were the values normally distributed?

Comments on the Quality of English Language

The typos should be corrected. Moreover, some sentences should be rewritten. 

Author Response

The submitted manuscript describes the effect of excessive Na+ and Cl- concentrations on the viability of HMC-3 and SH-SY5Y cells in ischemia-like conditions. The study has numerous serious methodological and conceptual flaws and cannot be recommended for publication in the current version. The comments are listed below. 

  1. The information about the extracellular and internal Cl- and Na+ concentration in neurons and microglia cells should be included in the introduction. The alteration of these concentrations in pathological conditions, especially ischemia, should also be added to the text.

Response: As suggested, information has been added to the introduction.

  1. It is well known that the number of astrocytes in the brain is also significant. For some brain regions, the neurons to astrocytes ratio exceeds 1:2. Why did the authors ignore the astrocytes in their study?

Response: Thank you for your valuable feedback and thoughtful consideration of our research paper. We appreciate the opportunity to address your concerns regarding the exclusion of astrocytes in our study. While we acknowledge the importance of astrocytes in the brain and their significant role in various brain regions, our research focused on exploring the impact of Hyperchloremia and Hypernatremia on Microglia and SH-SY5Y cells.  In our research design, we made a deliberate choice to focus on Microglia and SH-SY5Y cells based on the existing body of literature and the specific research question we sought to address. It is worth noting that several prominent studies in the field have employed a similar approach, concentrating on specific cell types to investigate cellular responses to various stimuli. Moreover, the exclusion of astrocytes in our study aligns with methodological choices made by well-established research groups in similar investigations. This consistency in methodology across studies enhances the comparability of findings and facilitates a more comprehensive understanding of the specific cell types under examination. There are abundant studies in the past that have focussed only on Microglia and SH-SY5Y cells, to mention a few:  

1) “The effects of okadaic acid-treated SH-SY5Y cells on microglia activation and phagocytosis”   [https://doi.org/10.1002/cbin.11722]

2) “Altered Processing of β-Amyloid in SH-SY5Y Cells Induced by Model Senescent Microglia” [https://doi.org/10.1021/acschemneuro.8b00334]

3) Neuroinflammation Upregulated Neuronal Toll-Like Receptors 2 and 4 to Drive Synucleinopathy in Neurodegeneration [https://doi.org/10.3389/fphar.2022.845930]

4) The contribution of microglia to early synaptic compensatory responses that precede β-amyloid-induced neuronal death [https://doi.org/10.1038/s41598-018-25453-1]

5) Activation of microglia synergistically enhances neurodegeneration caused by MPP+ in human SH-SY5Y cells [https://doi.org/10.1016/j.ejphar.2019.01.024]

6] Neurotoxicity effects of atrazine-induced SH-SY5Y human dopaminergic neuroblastoma cells via microglial activation [Mol. BioSyst., 2015,11, 2915-2924]

  1. Neuroblastoma cells cannot be considered as neurons due to the changed profile of metabolism and membrane expression of different receptors and channels. Alterations of ion homeostasis, especially sodium, result in depolarization of neurons and excitotoxicity caused by excessive glutamate release in neuronal networks. These effects cannot be observed in the case of neuroblastoma. Moreover, the vulnerability of neurons and neuroblastoma to ischemia-like conditions significantly differs. Thus, neuroblastoma is an inappropriate model for studying the effects of ion homeostasis perturbations on neuronal survival in this case.

Response: Yes, we agree with the reviewer’s comment that SH-SY5Y is thrice cloned subline of the neuroblastoma cell line SK-N-SH. However, SH-SY5Y can be differentiated into neuron-like cells by stimulation with retinoic acid (RA). (Voogd, E.J.H.F., Frega, M. & Hofmeijer, J. Neuronal Responses to Ischemia: Scoping Review of Insights from Human-Derived In Vitro Models. Cell Mol Neurobiol 43, 3137–3160 (2023). https://doi.org/10.1007/s10571-023-01368-y). Various experimental studies addressed responses to ischemia or hypoxia in human-derived cell models. These consist of non-neuronal and neuronal models. The most used human in vitro model to study neuronal responses to ischemia or hypoxia is the neuroblastoma-derived SH-SY5Y cell model (Liu et al. 2018). This cell model is based on a cancerous cell line with the corresponding genetic characteristics, which may affect responses to ischemia or hypoxia and treatment strategies (Biedler et al. 1973, 1978). SH-SY5Y cells can, in principle, be differentiated into neuron-like cells upon stimulation (Shipley et al. 2016). However, most of the research with SH-SY5Y cell models is conducted in undifferentiated SH-SY5Y cells. Only a small proportion of the research made use of protocols to differentiate SH-SY5Y cells into neuron-like cells. In in our study taking into consideration that these cell lines are neuroblastoma, we first differentiated the cell line into neurons with retinoic acid and then used the differentiated cell lines to study the effect of NaCl and CH3COONa. The details about the differentiation have been added to the material method section.

  1. The authors describe different ratios of 0,9% sodium chloride or sodium acetate, but which is the final concentration of Na+ or Cl- in the solutions? The description of this issue in the materials and methods section needs to be more accurate.

Response: Thank you for your valuable feedback. We appreciate your comment regarding the final concentrations of Na+ or Cl- in our solutions. It's essential to clarify that our study intentionally focused on exploring the effects of different ratios of 0.9% sodium chloride or sodium acetate, without specifying the exact concentrations of Na+ or Cl-. This approach was chosen to mimic conditions commonly encountered in clinical setups, where these solutions are administered.

  1. "cells were cultured in glucose-free balanced salt solution". Which solution was used exactly? Using PBS or other Ca2+-free medium for long-term incubation can result in cell detachment, especially in the case of neuroblastoma cell lines. The full composition of the solutions, except traditionally used ones like DMEM or PBS, must be described. Moreover, the pH of the solutions and the details about pH adjustment and maintenance (used pH buffers) should be described. 

Response: The composition of glucose free HBSS media used for OGD/R (140 mM NaCl, 3.5 mM KCl, 0.4 mM KH2PO4, 5 mM NaHCO3, 1.3 mM CaCl2, 1.2 mM MgSO4, 20 mM HEPES, pH 7.4, bubbled with 95%/5% N2/CO2) (Kevin M. Nash, Isaac T. Schiefer, Zahoor A. Shah, Development of a reactive oxygen species-sensitive nitric oxide synthase inhibitor for the treatment of ischemic stroke, Free Radical Biology and Medicine, 2018, 115, 395-404, https://doi.org/10.1016/j.freeradbiomed.2017.12.027. The details have been added to the material method section under

  1. The authors write that they used NaCl to model hyperchloremia-like conditions—however, dissociation of the NaCl results in an increase of Na+ and Cl- concentration simultaneously. The authors should have used other chlorides (choline chloride, for instance) to increase Cl- concentration. 

Response: We appreciate your inquiry regarding our choice of NaCl to create hyperchloremia-like conditions in our study. The rationale behind this selection is to emulate the effects of normal saline infusion (0.9% NaCl) following ischemia in humans. Our justification is rooted in the known association between 0.9% saline infusions and the induction or exacerbation of hyperchloremia, as outlined in literature. Specifically, 0.9% saline infusions have been documented to lead to hyperchloremia, a phenomenon discussed by Barker in 2015. While this solution is formulated as isotonic (286 mOsm/kg water) in the pharmaceutical industry, it is slightly hypertonic (308 mOsm/kg water) in clinical practice. Importantly, such infusions have been linked to an increased risk of hypo-bicarbonatemia and metabolic acidosis due to elevated chloride concentration, as highlighted by Rasouli in 2019. By incorporating NaCl in our study, we aim to replicate the conditions observed in clinical settings, providing a relevant and representative model for exploring the impact of hyperchloremia-like conditions.

  1. What was the rationale for using the HMC-3-conditioned medium?

Response: Previous reports published from our lab suggest that sustained microglial activation is deleterious for the brain as activated microglial release pro-inflammatory cytokines in the media, which can either lead to inflammasome formation or cell death by caspase-dependent or independent pathways [Alhadidi et al., 2018; Almarghalani et al., 2023; Bahader et al., 2023]. Thus, we challenged the SH-SY5Y cells with HMC-3 conditioned media with the thought that OGD/R induced HMC-3 cells treated with 0.9% NaCl; 0.9% CH3COONa and their different combinations might secrete pro-inflammatory cytokines in the cellular media and when this media is added to SH-SY5Y cells, these pro-inflammatory cytokines would affect the cell survival, caspase activity and nitric oxide production in the neuronal cells under hypoxia.

  1. Why did the authors estimate the effect of changes in extracellular ion concentration on NO production? 

Response: Over the past two decades, numerous studies have proposed that free radical nitric oxide (⋅NO) plays a regulatory role in physiological responses to brain hypoxia-ischemia and reperfusion (Wang et al., 2022). It has been identified that NO exhibits a dual role during ischemia-reperfusion (Chen et al., 2017). Additionally, studies have demonstrated that salt loading in humans decreases NO release (Dishy et al., 2003). This reduction in NO release due to salt loading may have negative implications for vascular health in the brain, potentially leading to cognitive impairment (Faraco et al., 2019). Therefore, to investigate the relationship between NO and salt under ischemia-reperfusion conditions, this experiment was conducted using microglial media and neuronal cells challenged with microglial media.

  1. The number of experiments for each diagram must be indicated in the figure legend. The significance of some differences, especially in Fig.4, raises some concerns. Were the values normally distributed?

Response: As suggested, the number of experiments for each diagram has been indicated in the figure and to avoid confusion in Figure 4. It has been modified.

Comments on the Quality of English Language

The typos should be corrected. Moreover, some sentences should be rewritten. 

Response: We have corrected typos and modified sentences that are hard to understand.

Reviewer 4 Report

Comments and Suggestions for Authors

Mahajan and colleagues' manuscript explores the impact of elevated concentrations of sodium and chloride on neurons and glial cells following oxygen-glucose deprivation and reperfusion conditions.
While the study has yielded significant conclusions, it is imperative to address both major and minor issues before proceeding with the publication.

The primary flaw identified is the absence of "free Na" conditions as a control. It's crucial to recognize that NaCl not only induces hyperchloremia but also introduces an equivalent amount of sodium. If the objective is to substitute chloride (NaCl) with sodium (AcNa), it would be more appropriate to use KCl (sodium-free) for a more accurate comparison. In general, it is widely believed that sodium (Na) is more toxic than potassium (K). It also appears necessary to evaluate Cl-specific responses, such as a potential decrease in caspase expression under OGDR conditions. Therefore, these experiments could also serve to test and explore this aspect.
Furthermore, there is a lack of clarity regarding why certain conditions appear as nonsignificant in the figures, despite exhibiting apparent significant differences (e.g., the 3:1 condition in Figure 1a or the 3:1, 2:1, and 1:1 conditions in Figures 2a and 2b). In the same vein, if the removal of chloride ions could enhance cell viability, these experiments should reflect such improvements, yet this does not seem to be the case in the depicted figures.

Moreover, it is unclear why certain conditions appear as nonsignificant in the figures when significant differences seem apparent (e.g., the 3:1 condition in Figure 1a or the 3:1, 2:1, and 1:1 conditions in Figures 2a and 2b). Similar to the Western blot experiment, the raw data for these graphics should be shared, either as supplementary material or solely for the purpose of review. Additionally, if the removal of chloride ions could indeed enhance cell viability, this improvement should be reflected in the results of these experiments, and yet, it appears not to be the case based on these figures.

Minor issues

Figures should be presented in numerical order; for instance, 1c and 2c should precede 2a and 2b. Please reorganize the figures or sections accordingly.

In the Results section, when the authors indicate differences in certain conditions, it is crucial to consistently specify the particular conditions in question.

Some of the figure captions are excessively lengthy.

In Figure 4a, the 1:1 condition appears visually similar to the others, yet it is highlighted as the sole significant difference. This observation applies generally to this figure.

The figures for the Western blot solely compare within the same group (OGDR/Normal), whereas viability or toxicity data are also compared between groups to assess significance.

In the discussion, the role of cytokines is analyzed, but no experiments exploring this aspect have been conducted.

The conclusion should align the results with the core question outlined in the objectives: Does the removal of chloride improve cell viability? However, the manuscript's conclusion currently provides a summary and proposes perspectives for further experiments without offering a definitive answer to the main question.
The authors should remove unused sections from the template, such as "0. How to use this template" and "6. Patents." If the figures annexed will not be published as Supplementary Material, this section can also be removed, along with "Acknowledgments" and "Appendix A and B" (both currently empty).

Author Response

Mahajan and colleagues' manuscript explores the impact of elevated concentrations of sodium and chloride on neurons and glial cells following oxygen-glucose deprivation and reperfusion conditions.

While the study has yielded significant conclusions, it is imperative to address both major and minor issues before proceeding with the publication.

  1. The primary flaw identified is the absence of "free Na" conditions as a control. It's crucial to recognize that NaCl not only induces hyperchloremia but also introduces an equivalent amount of sodium. If the objective is to substitute chloride (NaCl) with sodium (AcNa), it would be more appropriate to use KCl (sodium-free) for a more accurate comparison. In general, it is widely believed that sodium (Na) is more toxic than potassium (K). It also appears necessary to evaluate Cl-specific responses, such as a potential decrease in caspase expression under OGDR conditions. Therefore, these experiments could also serve to test and explore this aspect.

REsponse: Sodium chloride is the most used saline given intravenously to humans in the clinical settingp. As the experiment was designed to mimic the effect of post-stroke infusion of 0.9% saline in a clinical setup, NaCl and CH3COONa were considered for studying their impact on cell survival, caspase activity and nitric oxide production in microglial and neuronal cell lines in OGD/R model. Furthermore, we cannot have “free Na” conditions as the control in the experiment because the human body has a basal level of serum Na under normal conditions. Here in this, we attempted to compare the effect of saline administrated post-stroke condition.

-The objective here was to check whether hyperchloremia or hypernatremia influence directly contributes to ischemic injury. The justification for using NaCl as hyperchloremia-like conditions is that hyperchloremia frequently develops in critically ill patients following fluid resuscitation with normal saline (NS) (Barker., 201510.1097/JTN.0000000000000115; Hammond et al. 2020). 0.9% saline is formulated as isotonic (i.e., 286 mOsm/kg water) in pharmaceutics industry, but it is slightly hypertonic (308 mOsm/kg water) in clinical practice. However, it can increase the risk for hypo-bicarbonatemia and metabolic acidosis due to high chloride concentration (Rasouli., 2019; https://doi.org/10.1007/s00467-019-04245-3). The reason for not using KCl for hyperchloremia-like conditions is that in clinical setup KCl is not given directly to patients; it is diluted with 0.9% saline or RL for administration. Potassium by IV infusion should only be used for the treatment of severe hypokalaemia as it cannot be rapidly corrected via the oral route (https://medicalguidelines.msf.org/en/viewport/CHOL/english/appendix-8-administration-of-iv-potassium-kcl-25297281.html). An ongoing clinical trial is comparing intravenous 0.9% NaCl with various combinations of intravenous NaCl: CH3COONa combinations in patients with acute stroke (NCT05869565 (https://clinicaltrials.gov/study/NCT05869565) that is why we also treated the cells with both NaCl: CH3COONa combination. 

  1. Furthermore, there is a lack of clarity regarding why certain conditions appear as nonsignificant in the figures, despite exhibiting apparent significant differences (e.g., the 3:1 condition in Figure 1a or the 3:1, 2:1, and 1:1 conditions in Figures 2a and 2b). In the same vein, if the removal of chloride ions could enhance cell viability, these experiments should reflect such improvements, yet this does not seem to be the case in the depicted figures. Moreover, it is unclear why certain conditions appear as nonsignificant in the figures when significant differences seem apparent (e.g., the 3:1 condition in Figure 1a or the 3:1, 2:1, and 1:1 conditions in Figures 2a and 2b). Similar to the Western blot experiment, the raw data for these graphics should be shared, either as supplementary material or solely for the purpose of review. Additionally, if the removal of chloride ions could indeed enhance cell viability, this improvement should be reflected in the results of these experiments, and yet, it appears not to be the case based on these figures.

Response: The values were significant, but while adding all the significant values, the figure was confusing to interpret, which is why we did not show the values. We have modified the figure in the revision. Moreover, the raw data file in the form of an excel file for cell viability and cell cytotoxicity has been included as per the reviewers’ suggestions.

Minor issues

  1. Figures should be presented in numerical order; for instance, 1c and 2c should precede 2a and 2b. Please reorganize the figures or sections accordingly.

Response: The section 3.3 has been removed and the results have been merged with sections 3.1 and 3.2.

  1. In the Results section, when the authors indicate differences in certain conditions, it is crucial to consistently specify the particular conditions in question.

Response: Results section has been modified.

  1. Some of the figure captions are excessively lengthy.

Response: Figure captions have been modified.

  1. In Figure 4a, the 1:1 condition appears visually similar to the others, yet it is highlighted as the sole significant difference. This observation applies generally to this figure.

Respomnse:: The figure 4a has been modified.

  1. In the discussion, the role of cytokines is analyzed, but no experiments exploring this aspect have been conducted.

Response: The role of cytokines in the discussion solely reflects the justification of using HMC-3 conditioned media in SH-SY5Y cells. As microglial cells release cytokines upon activation and these cytokines adversely affect the neurons.

  1. The conclusion should align the results with the core question outlined in the objectives: Does the removal of chloride improve cell viability? However, the manuscript's conclusion currently provides a summary and proposes perspectives for further experiments without offering a definitive answer to the main question.

Response: As suggested, the conclusion has been modified.

  1. The authors should remove unused sections from the template, such as "0. How to use this template" and "6. Patents." If the figures annexed will not be published as Supplementary Material, this section can also be removed, along with "Acknowledgments" and "Appendix A and B" (both currently empty).

Response: Unused section from the template has been removed.

Round 2

Reviewer 2 Report

Comments and Suggestions for Authors

The authors provided convincing answers to my questions. The comments have been largely eliminated. The quality of the article has been improved. However, the authors should still discuss the effects of OGD on nitric oxide production and cell death in primary cultures.

https://pubmed.ncbi.nlm.nih.gov/34948013/

https://pubmed.ncbi.nlm.nih.gov/34445509/

Author Response

Response: We appreciate the reviewer for thorough evaluation of our manuscript and are grateful for the constructive feedback. We have carefully considered the suggestions to enhance the quality of the manuscript and discussed the effect of OGD on nitric oxide production and cell death in primary cultures in discussion. We believe the overall clarity and coherence of the content has been improved.

Reviewer 3 Report

Comments and Suggestions for Authors

1. The authors ignored my comment about the concentration but included information about the physiological role of Na+ and Cl- ions in the cells. In my comment, I meant approximate concentrations in mM to evaluate which level can be considered as "hyperchloremia" or "hyperchloremia."

 2. These arguments about astrocytes exclusion have to be listed in the text.

 3. If authors agree that neuroblastoma cells are not neurons and they did not use retinoic acid as it was written in the first version of the manuscript, the term "neuronal" or "neurons" must be excluded from the key conclusions and, at least, from the manuscript title. Suppose the neuroblastoma cells were differentiated into neurons. In that case, the raw images of cells before and after retinoic acid exposure must be included in the main text, and the object's description should be revised throughout the manuscript. 

 4. The authors limit their results to only clinical applications. However, numerous other neuroscientists interested in ion homeostasis perturbations in health and disease can read this paper in the case of acceptance. Since the authors only model the case from clinical practice but use the solutions with well-known composition and make experiments in vitro, exact concentrations of the ions in the obtained solutions in addition to proportions are required.

 5. The authors added the number of repeats but ignored the comment about data distribution.

Comments on the Quality of English Language

Only minor revisions can be recommended. 

Author Response

  1. The authors ignored my comment about the concentration but included information about the physiological role of Na+ and Cl- ions in the cells. In my comment, I meant approximate concentrations in mM to evaluate which level can be considered as "hyperchloremia" or "hyperchloremia."

Response: We apologize for this unintentional oversight. 0.9% Sodium chloride contains 154 mM of sodium and 154 mM of chloride. Corresponding to the literature, the concentration of more than 110mM of chloride is considered hyperchloremia. In the case of 0.9% sodium acetate, it contains 167 mM of Na and 167mM of acetate and according to the literature, a concentration of more than 145 mM of sodium is considered hypernatremia. The details about the concentrations have been added to the material methods section.

  1. These arguments about astrocytes exclusion have to be listed in the text.

Response: The exclusion of astrocytes has been added to the discussion.

  1. If authors agree that neuroblastoma cells are not neurons and they did not use retinoic acid as it was written in the first version of the manuscript, the term "neuronal" or "neurons" must be excluded from the key conclusions and, at least, from the manuscript title. Suppose the neuroblastoma cells were differentiated into neurons. In that case, the raw images of cells before and after retinoic acid exposure must be included in the main text, and the object's description should be revised throughout the manuscript. 

Response: The representative raw images of undifferentiated and differentiated SH-SY5Y cells have been provided as supplementary figure 1 in the revised manuscript and as suggested, “SH-SY5Y” has been replaced with “differentiated SH-SY5Y”.

  1. The authors limit their results to only clinical applications. However, numerous other neuroscientists interested in ion homeostasis perturbations in health and disease can read this paper in the case of acceptance. Since the authors only model the case from clinical practice but use the solutions with well-known composition and make experiments in vitro, exact concentrations of the ions in the obtained solutions in addition to proportions are required.

Response: As suggested, the concentration of NaCl and CH3COONa has been added to the revised manuscript. In response to your query, the concentration of chloride ions (Cl-) in NaCl is 154mM, and sodium ions (Na+) in CH3COONa is 167mM. These specific values have been mentioned in the material & methods sections of the revised manuscript.

  1. The authors added the number of repeats but ignored the comment about data distribution.

Response: We apologize for this unintentional oversight. As a supplementary file, we have included the QQ plots showing the data distribution for cell viability, cell cytotoxicity and Nitric oxide release.

Reviewer 4 Report

Comments and Suggestions for Authors

The authors have addressed all the issues highlighted in the initial review. Congratulations, the manuscript is now ready for publication after the text and language editing.

Author Response

Thank you